

# Non-destructive environmental safety assessment of threatened and endangered plants in weed biological control

Ikju Park[1,2], Mark Schwarzländer[2], Sanford D. Eigenbrode[2], Bradley L. Harmon[2], Hariet L. Hinz[3] and Urs Schaffner[3]

[1] Department of Entomology, University of California, Riverside, Riverside, California, United States
[2] Department of Entomology, Plant Pathology and Nematology, University of Idaho, Moscow, Idaho, United States
[3] CABI, Delémont, Switzerland

Corresponding author
Ikju Park, ikju.park@ucr.edu

## ABSTRACT

Assessing the risk of nontarget attack (NTA) for federally listed threatened and endangered (T&E) plant species confamilial to invasive plants targeted for classical biological control, is one of the most important objectives of pre-release environmental safety assessments in the United States. However, evaluating potential NTA on T&E species is often complicated by restrictive agency requirements for obtaining propagules, or the ability to propagate plants and rear agents to the appropriate phenostages synchronously for testing, or both. Here, we assessed whether plant cues associated with a host recognition can be used for testing the attractiveness of four T&E and one rare single population plant species non-destructively for a candidate biocontrol agent. We used the seed-feeding weevil, *Mogulones borraginis*, a candidate for the biological control of the invasive plant, *Cynoglossum officinale* (Boraginaceae) as the study system. We collected olfactory and visual cues in the form of flowering sprigs from T&E plant species confamilial to the invasive plant in a non-destructive manner and used them to measure behavioral responses and searching time of weevils. Female weevils preferred *C. officinale* to all tested plant species in dual-choice bioassays using either olfactory or visual cues in a modified y-tube device. Furthermore, female weevils were repelled by the combined olfactory and visual cues from all tested T&E plant species in a dual-choice test against controls (*e.g.*, purified air in an empty arm), indicating that it would be extremely unlikely for the weevil to attack any of these species upon release in the United States. Principal component analysis based on 61 volatile organic compounds effectively separated the five confamilial plant species and *C. officinale*, corroborating the results of behavioral bioassays. We conclude that studies on pre-alighting host selection behavior and the underlying physiological mechanisms of how organisms select host plants they exploit can aid in environmental safety testing of weed biological control agents.

## INTRODUCTION

In classical biological control of weeds, pre-release risk assessment studies are essential to predict whether a biological control candidate will attack nontarget native plant species (*Grevstad, McEvoy & Coombs, 2021*; *Hinz, Winston & Schwarzländer, 2019*; *Paynter, Paterson & Kwong, 2020*; *Schwarzländer et al., 2018*). For the United States, concerns about nontarget attack (NTA) are greatest for plant species related to the target weed and listed as either threatened or endangered (*Ancheta & Heard, 2011*; *Shirey et al., 2013*). NTA by classical biological control agents on native threatened and endangered (T&E) plant species in the United States have prompted concerns about the reliability of pre-release risk assessment protocols (*Gijsman, Havens & Vitt, 2020*; *Havens et al., 2012*; *Louda et al., 2003*; *Rand, Russell & Louda, 2004*; *Stiling, Moon & Gordon, 2004*; *Strong, 1997*). While recent reviews have concluded that less than 1% of released weed biological control agents have potentially caused population level NTA (*Hinz, Winston & Schwarzländer, 2020*; *Suckling & Sforza, 2014*), there is still a need to develop new protocols that can further enhance environmental safety predictions in weed biological control (*Hinz, Winston & Schwarzländer, 2019*). Another recent review on outcomes of weed biological control programs argued that additional NTA may go undetected or unreported (*Havens et al., 2019*). One approach to improve pre-release assessment of potential NTA is to include ecological information such as the host-finding behavior of biological control candidate species (*Heard, 2000*; *Knolhoff & Heckel, 2014*; *Louda et al., 2003*; *Zwölfer & Harris, 1971*). The argument being that it is unlikely that a plant species would be attacked post-release if the biological control agent is unable to recognize it as a host plant, even if the nontarget species was attacked in confined cage trials. It would be particularly useful if host-finding studies could be conducted as non-destructively as possible (without any disturbance to natural populations) since it would allow the inclusion of threatened or endangered (T&E) plant species related to the invasive plant species targeted for biological control (*Minteer et al., 2020*; *Fung et al., 2022*).

Two matters complicate the assessment of NTA risk for T&E plant species in weed biological control in the United States. First, permits to collect or acquire propagules of T&E plant species and to move them across states are strictly regulated by the US Fish and Wildlife Service and other federal and state agencies (*Shirey et al., 2013*). Second, if permits are granted and propagules received, it is often difficult to grow these species to the phenostage necessary for conducting meaningful host specificity tests (*e.g.*, fruiting for a seed-feeding biological control candidate). Thus, surrogates of T&E plant species (*i.e.*, congeners with similar phenotypic characters and overlapping distribution) are often used for testing the pre-release risk assessment of potential biological control candidates.

Insect herbivores evaluate plant cues of host and nonhost plants in the pre-alighting phase (*Miller & Strickler, 1984*). Olfaction and vision are the two plant cue modalities active during the pre-alighting stage of host finding (*Bernays & Chapman, 1994*; *Kennedy, 1978*; *Prokopy, 1986*), which can determine whether or not a plant is a potential host (*Clement & Cristofaro, 1995*; *Heard, 2000*; *Marohasy, 1998*; *Park et al., 2018*; *Schaffner, Smith & Cristofaro, 2018*; *Schiestl, 2015*; *Wheeler & Schaffner, 2013*). Evaluating the

behavior of candidate agents during the pre-alighting phase of host selection can strengthen environmental safety assessments of weed biological control agents (*Andreas et al., 2009*; *Fung et al., 2022*; *Müller & Nentwig, 2011*; *Park, Schwarzländer & Eigenbrode, 2011*; *Park et al., 2019*; *Sutton et al., 2017*). A simultaneous evaluation of visual and olfactory cues has rarely been attempted to determine the environmental safety of weed biological control candidates in general and never for T&E plant species confamilial to the targeted invasive plant (but see *Park & Thompson (2021)* for testing olfactory cues from both T&E and invasive thistles).

The purpose of this study was to test the attractiveness of plant species during the pre-alighting phase for non-destructive environmental safety assessment of federally listed T&E plant species in weed biological control. We used a biological control candidate, the seed-feeding weevil, *Mogulones borraginis* F. (Coleoptera: Curculionidae), the plant *Cynoglossum officinale* L. (Boraginaceae) which is invasive in North America, and five closely related plant species (four T&E species and one imperiled species) native to the United States as a model system. We used methods pioneered using *M. borraginis* to assess its responses to visual and olfactory cues *of C. officinale* and three non-T&E plants as a proof of concept (*Park et al., 2018*). Thus, this study aimed to assess whether testing *M. borraginis* attraction to olfactory and visual cues offers an opportunity for non-destructive assessment of the likelihood of NTA of the most critical test plant species, *i.e.*, of T&E species closely related to the target weed.

## MATERIALS AND METHODS

### Insects and plants

We received naïve overwintering *M. borraginis* from a rearing colony at CABI in Switzerland typically in early May of each year of this 4-year study. Because female *M. borraginis* must feed on flowers of *C. officinale* to initiate oogenesis, we provided fresh cymes of *C. officinale* to weevils and kept them in an environmental chamber (E-30B, Percival Scientific, Perry, IA, USA; L18: D6 at 20 °C and 60% relative humidity) at the University of Idaho weed biological control quarantine laboratory. Since T&E plant species under the Endangered Species Act were included on the test plant list for the pre-release risk assessment of *M. borraginis*, four T&E and one imperiled plant species in Boraginaceae were selected in the United States using royalty-free map software (www.mapsfordesign.com) (Fig. 1). Seeds of *Amsinckia grandiflora* (Kleeb. Ex A. Gray) Kleeb. Ex Greene ($n = 100$), *Plagiobothrys strictus* (Greene) I.M. Johnst. ($n = 100$), *Plagiobothrys hirtus* (Greene) I.M. Johnst. ($n = 100$), and *Hackelia venusta* (Piper) H. St. John ($n = 50$) were received from collaborators (see the acknowledgements). They were sown in Sunshine Mix #2 (Sun Gro Horticulture Canada Ltd., Vancouver, Canada) at 4 °C for 12 weeks. Another T&E plant species, *Oreocarya crassipes* (I. M. Johnst.) Hasenstab & M.G. Simpson, was excluded because it is adapted to an arid environment that both *C. officinale* and *M. borraginis* could not tolerate. Rootstocks of *Dasynotus daubenmirei* I.M. Johnston ($n = 10$) were collected at Walde Lookout, ID, USA (N 46.23528°,

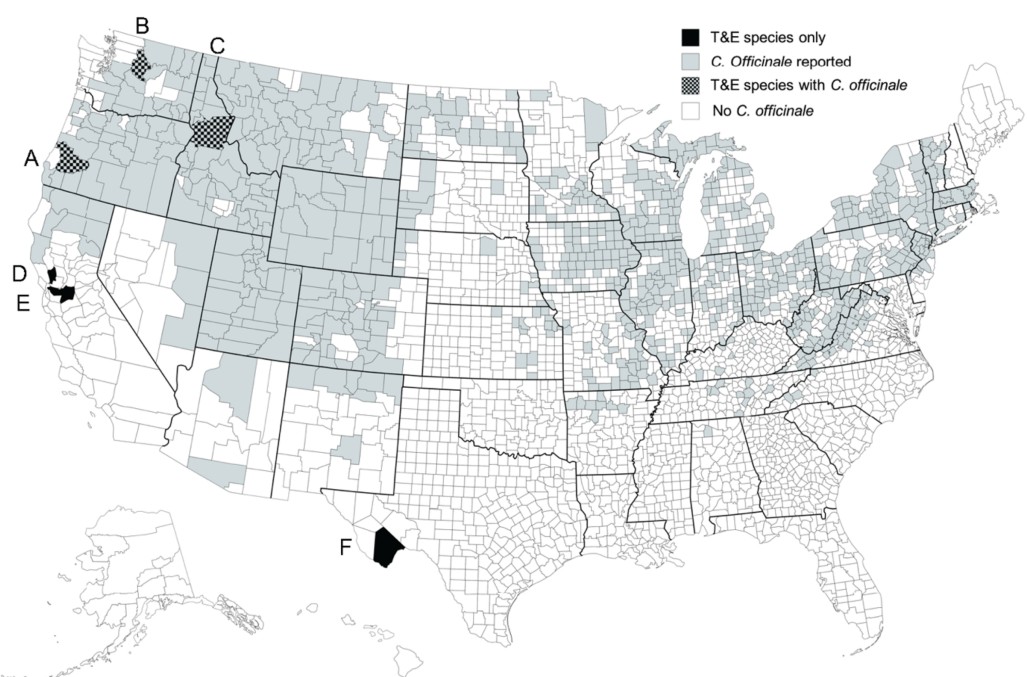

**Figure 1 Distribution of T&E plant species in the U.S.** Locations of remaining field populations of the five federally listed threatened and endangered (T&E) plant species and one rare single-population species in the Boraginaceae family in the United States. (A) *Plagiobothrys hirtus*, (B) *Hackelia venusta*, (C) *Dasynotus daubenmirei*, (D) *Plagiobothrys strictus*, (E) *Amsinckia grandiflora*, and (F) *Oreocarya crassipes*. The map was produced using royalty-free map software (www.mapsfordesign.com).

W 115.63528°). Although *D. daubenmirei* is not a federally listed T&E species, it is the single species in its genus and grows exclusively in one population in north central Idaho (*Cohen, 2014*). Rootstocks of *C. officinale* (*n* = 20) were collected at the Idler's Rest Nature Preserve, Moscow, ID, USA (N 46.804160°, W 116.948554°). We transplanted them in 11.3 L black plastic pots filled with Sunshine Mix #2 (Sun Gro Horticulture, Agawam, MA, USA) and placed them in an environmentally-controlled greenhouse at the University of Idaho, Moscow, ID, USA in March of the following growing season. Plants were watered as needed. Because many species in Boraginaceae bloom along cymes containing buds, open inflorescences and young fruits, all of which are used by weevils as food and oviposition resource, we used plants bearing cymes for all experiments described below.

For *Amsinckia grandiflora*, *D. daubenmirei* and *P. hirtus*, single-choice oviposition test results were available (M. Schwarzländer, H.L. Hinz, R.L. Winston, 2020, unpublished data), and these species were used as "prove of concept" to study behavioral responses of a biological control agent to olfactory and visual plant cues in lieu of traditional host-specificity tests with potted plants. The assumption was that results of traditional tests would be confirmed and a physiological explanation for results could be provided. For *H. venusta* and *P. strictus* no propagules could be obtained. All plant cues for these two plant species were collected in the field and used in behavioral bioassays.

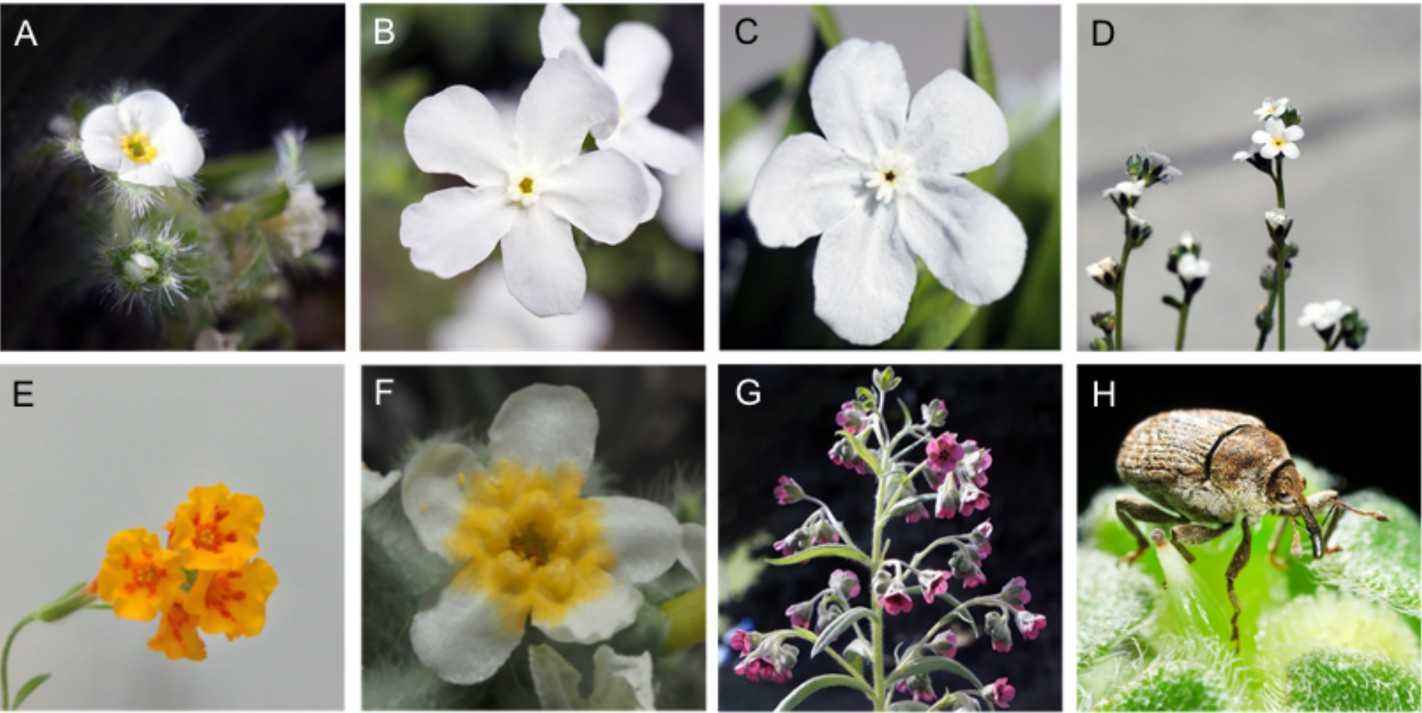

**Figure 2 Pictures of each plant species studied in this study.** Inflorescences of plant species included in this study (A–G). (A) *Plagiobothrys hirtus*, (B) *Hackelia venusta*, (C) *Dasynotus daubenmirei*, (D) *Plagiobothrys strictus*, (E) *Amsinckia grandiflora*, (F) *Plagiobothrys strictus*, (G) *Cynoglossum officinale*, and (H) *Mogulones borraginis*, the potential biological control agent for *C. officinale* (photo credit: I. Park).

## Collecting headspace volatile organic compounds (VOCs) and flowering stems

Polyvinyl acetate (PVA) bags (12 cm × 24 cm; Reynolds, Richmond, VA, USA) were purged of volatile contaminants in a drying oven at 140 °C for 60 min (*Park et al., 2018*). A flowering stem of the target weed and each test plant species was covered with the PVA bag and sealed with purged cotton balls and a cable tie to minimize potential physical damage to plants. Data were collected as previously described in *Park et al. (2018)*. Each elutant was placed in a screw cap vial and stored at 4 °C until further use. We collected VOCs of *C. officinale*, *H. venusta* and *P. hirtus* at their natural habitats, while VOCs from *Amsinckia grandiflora*, *D. daubenmirei*, and *P. strictus* were collected from plants grown in a greenhouse. In addition, permits were obtained to cut individual flowering stems (5 cm) of the respective three plant individuals per species and placed them into 10 cm transparent aqua-tubes (31-01931-case; Oasis Floral Products, Kent, OH, USA) and stored in a portable cooler to be used as visual cues in bioassays (Fig. 2). Further, to examining the influence of floral size and color on the host recognition by *M. borraginis*, we measured the size of petals and stamens of each plant species (*n* = 20, except for *P. strictus, n* = 10) using a digital caliper.

## Host-finding bioassays with double-stacked y-tube device (D-SYD)

A double-stacked y-tube device (D-SYD; diameter: 2 cm, length of the arms: 12 cm) was used to assess quantitatively the host selection behavior of *M. borraginis* based on olfactory (VOCs) and visual cues (a flowering stem) either individually or simultaneously (*Park et al., 2019*). Olfactory cues were offered to the insect through a pull pump in a closed system. Because olfactory and visual cues were separated by glass, there was no possibility that visual cues emitting VOCs could have contaminated the bioassays. The D-SYD was installed in a darkened room with a full spectrum light bulb (ES5M827FS; 27 Watts, Home Depot, Atlanta, GA, USA; 350 nm to 850 nm wavelength) diffused through a white polyethylene dome (40 cm × 30 cm × 20 cm) placed 20 cm above the D-SYD. The device was rinsed with 70% ethanol following each bioassay and rotated 180° after every five replicates to eliminate left or right-handed bias (*Park et al., 2018*). Twenty female weevils were used in each of the four experimental set ups per plant species (total of 80 female weevils for each plant species). Each female was only used once, and each individual weevil was treated as one replicate. For *P. strictus*, only ten females were used in each bioassay due to the limited number of *M. borraginis* available. In each replicate, the female was placed at the release point in the bottom Y-tube in each bioassay and observed for up to 5 min. A weevil was considered to have made a choice if it passed a decision line that was 3 cm into one of the arms within 5 min (*Tooker, Crumrin & Hanks, 2005*).

A total of four bioassays were conducted. Before conducting abovementioned experiments, neither visual nor olfactory cues were placed in the D-SYD and 30 female weevils tested, to confirm no bias exists (experiment 1). The next two experiments examined the role of a single cue of a T&E plant species and *C. officinale*. For the visual cue (experiment 2), 5 cm of a flowering stem was placed in the D-SYD; to test the effect of olfactory cues (experiment 3), 1 µl of headspace VOCs, which was eluted directly from the identical plants as in experiment 2, was placed in the D-SYD. The other two experiments evaluated the combined effect of visual and olfactory cues; the only difference was the presence (experiment 4) or absence (experiment 5) of *C. officinale* in addition to a plant species in the D-SYD.

## GC-MS and GC-EAD/FID analysis

Gas chromatography-mass spectrometry (GC-MS) was performed to check the identity of electrophysiologically active chemical compounds and quantifying their concentration in floral scents. An Agilent 7890A (Agilent Technologies Inc., Santa Clara, CA, USA), with an HP-5MS column (30 m × 250 µm × 0.25 µm; Agilent Technologies Inc., Santa Clara, CA, USA), was coupled with a Hewlett Packard 5973 mass selective detector (Agilent Technologies Inc., Santa Clara, CA, USA). A total of 10 ng of nonyl acetate (W278807, Sigma Aldrich) was injected with each elutant (1 µl) as an internal standard in the splitless mode, injector temperature 250 °C. Column specification and the temperature program were collected as previously described in *Park et al. (2019)*. To confirm the chirality of an enantiomeric compound, α-copaene, the floral blends were analyzed using an Agilent J&W Cyclodex-B column (Agilent Technologies Inc., Santa Clara, CA, USA). Retention time of the α-copaene was compared with the compound's retention times in two essential

oils (Young Living Essential Oils, Lehi, UT): *Angelica* in which (−) predominates and *Copaiba* in which (+) predominates.

Gas chromatography-electroantennographic detection and flame ionization detection (GC-EAD/FID) was performed to detect electrophysiologically active VOCs from *C. officinale*. If all female weevils (*n* = 6) responded to a VOC in the blend of *C. officinale*, the specific compound was considered as the electrophysiologically active VOC. An Agilent-6890N (Agilent Technologies, Santa Clara, CA) was equipped with an identical column specification and using the same temperature program as described in *Park et al. (2019)*. A 1:1 column splitter delivered effluent to the FID detector and to the EAD. Effluent was delivered *via* a GC effluent conditioner (Syntech, Hilversum, Netherlands) into humidified air flowing at 10 ml/sec directed through a glass tube to the antenna of a female weevil. Depolarization of the antenna was recorded and plotted with the FID signal using Syntech GC-EAD software.

To prepare antennae for recording, female *M. borraginis* were decapitated using a scalpel under a microscope. The decapitated head was placed on a ground probe with Spectra 360 electrode gel (Parker Laboratories, Fairfield, NJ, USA). The distal tip of the antenna was punctured with a minute insect pin (1208SA; Bioquip, Czech Republic), which was placed in contact with the recording probe (Ockenfels Syntech GmbH, Buchenbach, Germany). The undamaged antenna from the head was positioned to receive the entrained effluent from the GC column. The performance of the system was checked before each recording using an antenna simulator (Ockenfels Syntech GmbH, Buchenbach, Germany).

## Statistical analysis

We used a generalized linear model with an expected null ratio of 50:50 and a binominal distribution to analyze the number of weevils in the either side of testing device in each assay assuming a completely random design. We compared the least square means of responses to test whether behavioral responses of *M. borraginis* differed among control (experiment 1), single modalities (experiments 2 & 3), and combined cues (experiments 4 & 5) that compared the effect of absence/presence of *C. officinale* on weevil response in the two cue bioassays. We used a generalized linear model with a Poisson distribution to analyze the decision time (*i.e.*, the time delay from the initial movement of a female to passing the decision line) in each assay. Least squares mean differences were calculated to compare the decision time of females between arms of the D-SYD in each behavioral assay. Data from tests on all plants were pooled for analysis because there was no evidence of effects of individual plants on the weevil responses (*Park et al., 2018*). We performed two principal component analyses (PCA) based on the entire volatile profile of plants and the presence of electrophysiologically active VOCs identified from *C. officinale*. We compared the size of stamens and corollas among plant species with a one-way ANOVA. All analyses were carried out using SAS 9.4 (SAS Institute, Cary, NC, USA).

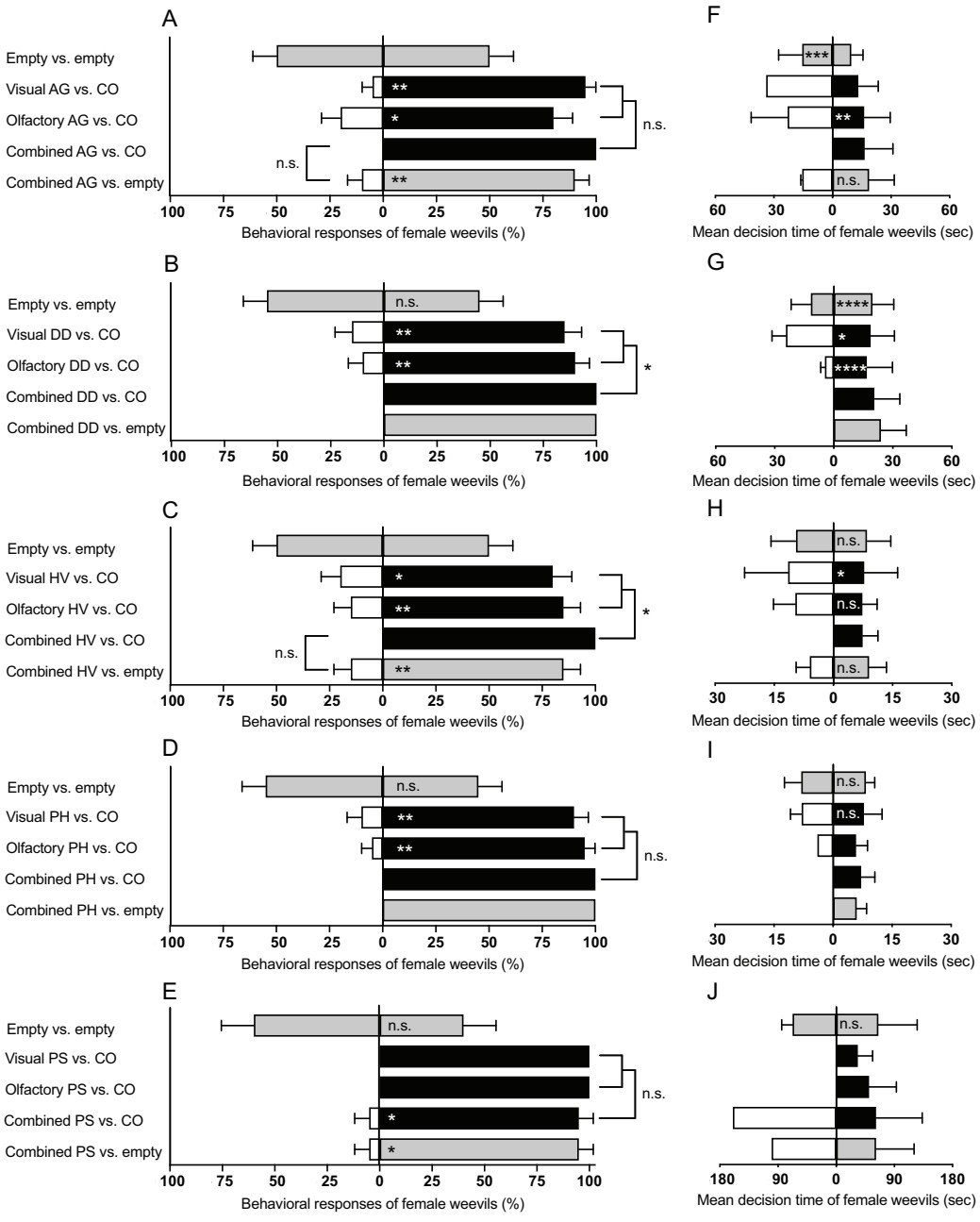

**Figure 3 Summary of behavioral bioassays on native plant species in this study.** Behavioral responses (A–E) and mean decision time (F–J) of female *M. borraginis* to olfactory, visual or combined olfactory and visual cues in dual choice bioassays in the D-SYD modified y-tube olfactometer. AG, *Amsinckia grandifora*; DD, *Dasynotus daubenmirei*; HV, *Hackelia venusta*; PH, *Plagiobothrys hirtus*; PS, *Plagiobothrys strictus;* CO = *C. officinale.* Grey fill = empty arm; black fill = cues from *C. officinale*; No fill = cues from T&E or imperiled plant species. Significance levels of the generalized linear model of individual assays (within bars) and comparisons between assays (brackets): n.s., not significant; *P < 0.05, **P < 0.01, ***P < 0.001, ****P < 0.0001.

## RESULTS

### Experiment 1: Empty (control) arms of D-SYD

The number of female weevils selecting either side of the empty D-SYD did not differ in all bioassays: (*A. grandiflora*: Z = 0, P = 1; *D. daubenmirei*: Z = 0.45, P = 0.65; *H. venusta*: Z = 0, P = 1; *P. hirtus*: Z = 0.45, P = 0.65; *P. strictus*: Z = 0.63, P = 0.52; Figs. 3A–3E, top bars in each panel). However, the searching time of the weevils in either side of the D-SYD differed for some experiments (Figs. 3F–3J, first bar from top). Time spent by female weevils on one side of the D-SYD was longer for *A. grandiflora* (Z = 3.69, P < 0.001; Fig. 3F) and *D. daubenmirei* (Z = 4.52, P < 0.0001; Fig. 3G), but not different for *H. venusta* (Z = 0.74, P = 0.45), *P. hirtus* (Z = −0.19, P = 0.85), and *P. strictus* (Z = 0.43, P = 0.66).

### Experiment 2: Visual cues

Female weevils preferred flowering stems of *C. officinale* over all plant species in this study (*A. grandiflora*: Z = 2.87, P < 0.01; *D. daubenmirei*: Z = 2.77, P < 0.01; *H. venusta*: Z = 2.48, P < 0.05; *P. hirtus*: Z = 2.95, P < 0.01; *P. strictus*: Z = 0, P = 1; Figs. 3A–3E, second bars from top). The decision time of female weevils did not differ for *P. hirtus* (Z = 0.08, P = 0.93), but was longer for *D. daubenmirei* (Z = 2.00, P = 0.04) and *H. venusta* (Z = 2.24, P = 0.02) when compared with *C. officinale* (Figs. 3F–3J, second bars from top). Only one female weevil chose *A. grandiflora* and *P. strictus* over *C. officinale*, which did not allow for statistical inferences of the decision time.

### Experiment 3: Olfactory cues

Female weevils preferred VOCs of *C. officinale* to all T&E plant species (*A. grandiflora*: Z = 2.48, P < 0.05; *D. daubenmirei*: Z = −2.95, P < 0.01; *H. venusta*: Z = 2.77, P < 0.01; *P. hirtus*: Z = 2.87, P < 0.01; *P. strictus*: Z = 0, P = 1; Figs. 3A–3E, third bars from top). Compared to *C. officinale*, the decision time of females did not differ for *H. venusta* (Z = 1.33, P = 0.18), was longer for *A. grandiflora* (Z = 2.86, P < 0.01) and was shorter for *D. daubenmirei* (Z = 3.90, P < 0.0001) (Figs. 3F–3H, third bars from top). The decision time for *P. hirtus* and *P. strictus* could not be statistically analyzed because only one weevil chose them over *C. officinale* (Figs. 3I–3J, third bars from top).

### Experiment 4: Combined visual and olfactory cues

When combining both cues, *M. borraginis* female strongly preferred *C. officinale* over all T&E plant species (*A. grandiflora*: Z = 0, P = 1; *D. daubenmirei*: Z = 0, P = 1; *H. venusta*: Z = 0, P = 1; *P. hirtus*: Z = 0, P = 1; *P. strictus*: Z = 2.08, P = 0.03; Figs. 3A–3E, fourth bars from top). Since no weevils chose any three T&E and one imperiled plant species, except one *M. borraginis* that chose *P. strictus*, we were unable to compare decision time between *C. officinale* and plant species (Figs. 3F–3J, fourth bars from top). Behavioral responses of *M. borraginis* did not differ when comparing response to a single plant cue with those to combined plant cues for three T&E plant species: *A. grandiflora* ($\chi^2$ = 3.28, P = 0.07), *P. hirtus* ($\chi^2$ = 2.34, P = 0.12), and *P. strictus* ($\chi^2$ = 2.27, P = 0.13) (Figs. 3A–3E, brackets to right of bars). The behavioral responses of *M. borraginis* were stronger for combined cues

than a single cue for *D. daubenmirei* ($\chi^2$ = 4.18, *P* = 0.04) and *H. venusta* ($\chi^2$ = 6.06, *P* < 0.05) (Figs. 3A–3E, brackets to the right).

**Experiment 5: Combined cues *vs.* control**
Female weevils strongly preferred the empty arm of the D-SYD to the combined visual and olfactory cues of all four T&E plant species and one imperiled plant species (*A. grandiflora*: Z = −2.95, *P* < 0.01; *D. daubenmirei*: Z = 0, *P* = 1; *H. venusta*: Z = 2.77, *P* < 0.01; *P. hirtus*: Z = 0, *P* = 1; *P. strictus*: Z = 2.08, *P* = 0.03; Figs. 3A–3E, bottom bars). The strength of this preference for combined cues was similar when the opposing arm of the D-SYD presented combined cues from *C. officinale* (Figs. 3A–3E; this comparison could not be made for *P. hirtus*, and *D. daubenmirei* because 100% of females chose *C. officinale*). The decision time of females did not differ for *A. grandiflora* (Z = −1.02, *P* = 0.30) and *H. venusta* (Z = −1.65, *P* = 0.09) (Figs. 3F–3J, bottom bars).

**Volatile profiles of plant species**
Sixty-one volatile compounds were identified in the floral scents of the six plant species included in this study. Of these, ten volatile compounds detected in the floral scent of *C. officinale* were electrophysiologically active based on EAD responses of *M. borraginis* females. Among them, two sesquiterpenes, (−)-α-copaene and (E)-β-farnesene, were unique to *C. officinale* when compared to the five T&E plant species. A PCA for VOC similarity among plant species explained 75.28% of variation (PC1: 41.57%, PC2: 33.72%) based on the ten electrophysiologically active VOCs found in *C. officinale* (Fig. 4A). When based on all 61 VOCs identified among the six plant species, the PCA explained 45.24% of the variation of the VOC similarity (PC1: 23.22%, PC2: 22.02%, Fig. 4B).

## DISCUSSION

The findings reported here show that *M. borraginis* strongly preferred cues from *C. officinale* over those of the T&E species tested, and that females even preferred pure air to the VOCs from these T&E species, effectively rejecting them as potential hosts. This is also supported by a separation of *C. officinale* to each plant species in the principal component analysis and provides a strong indication that in the field, the nontarget species we tested are very unlikely to be visited by female *M. borraginis* and might even be actively avoided prior to alightment. Importantly, this appears to hold for situations where the target weed, *C. officinale* may be present, or situations where it might be absent, creating a "no-choice" situation in the field. Even more importantly, these results were confirmed for three of the five species tested in single-choice oviposition tests, with zero eggs laid on *A. grandiflora*, *D. daubenmirei* and *P. hirtus* (M. Schwarzländer, H.L. Hinz, R.L. Winston, 2020, unpublished data). Although *H. venusta* could not be tested using traditional host-specificity methods, two other *Hackelia* species tested did not receive eggs either. It suggests that in the field, even in the absence of the target host, the nontarget species we tested are very unlikely to be visited by female *M. borraginis* and might even be actively avoided prior to alightment. The results of this study indicate that evaluating olfactory and visual cues can further improve the reliability of pre-release assessments to predict

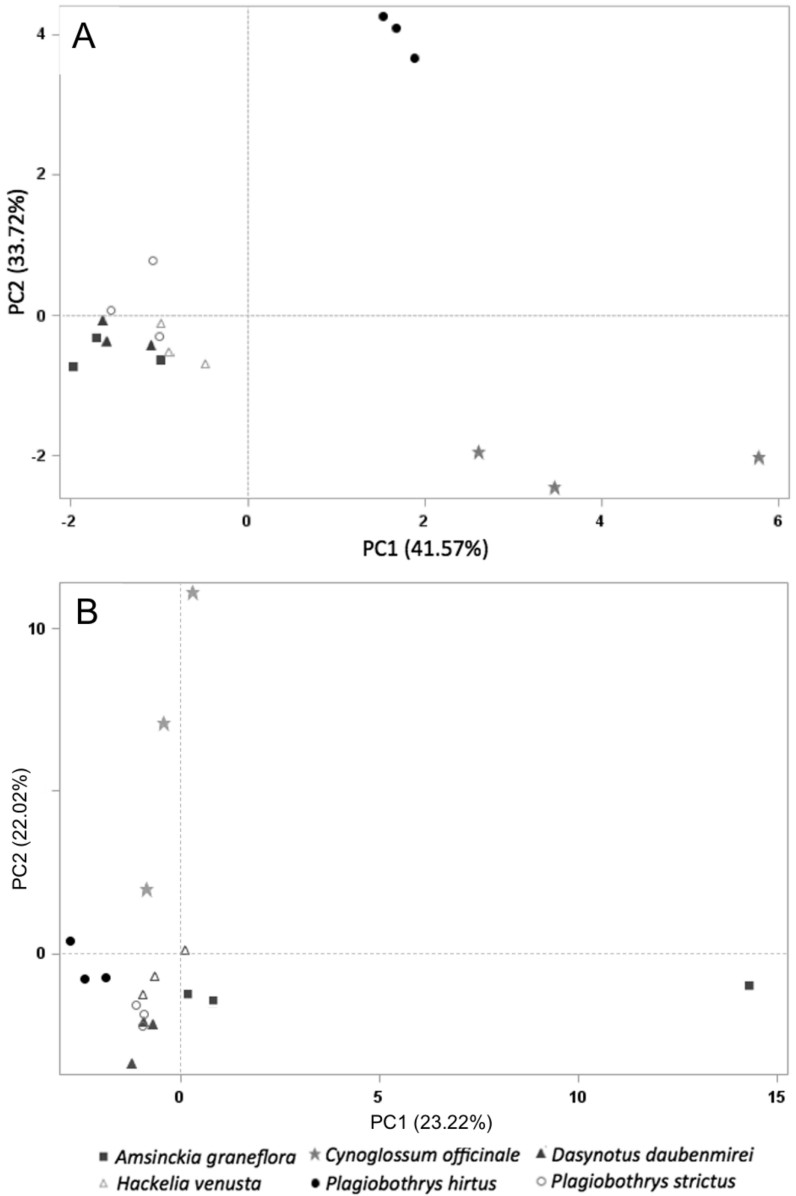

**Figure 4 PCA on all plant species tested in this study.** Principal component analysis based on the relative proportion of 61 chemical compounds in the volatile headspace blends of plant species used in the study (A) and 10 electrophsyiologically active chemical compounds (B): *A. grandiflora* (filled squares), *C. officinale* (filled stars), *D. daubenmirei* (filled triangles), *H. venusta* (open triangles), *P. hirtus* (filled circles) and *P. strictus* (open circles).

post-release nontarget attack on T&E plant species (*Hinz, Winston & Schwarzländer, 2019*; *Hinz, Winston & Schwarzländer, 2020*; *Park, Schwarzländer & Eigenbrode, 2011*; *Park et al., 2018, 2019*; *Park & Thompson, 2021*; *Paynter, Paterson & Kwong, 2020*).

In weed biological control, the inability to obtain T&E plant species for host specificity testing has traditionally been addressed by testing of surrogates of these species, *i.e.*, congeneric species that have similar habitat requirements and geographic ranges (*Colpetzer et al., 2004*; *Grevstad et al., 2013*; *Minteer et al., 2020*) and are presumed to be

comparable in their olfactory and visual plant cues. However, a review of the use of surrogate species shows that surrogates often do not share similar traits (*Che-Castaldo & Neel, 2012*). Our approach provides a better alternative to the use of surrogate species because it enables testing responses to T&E plant species of concern using volatile cues from individuals of these species growing in their natural habitat (*e.g.*, *H. venusta, P. hirtus*, and *D. daubenmirei*). Even when T&E plants cannot be sampled in their natural habitats, greenhouse-grown specimens can be used effectively with this protocol. We were able to show in a prior study that responses of *M. borraginis* to cues of *A. occidentale* did not differ between those collected from greenhouse-propagated plants or plants growing in their natural habitat (*Park et al., 2019*). In the present study, we were not able to obtain olfactory and visual cues of *P. strictus* and *A. grandiflora* in their natural habitat in California, using instead cues from greenhouse-propagated plants of these species.

The approach proposed here could allow to make a decision about the environmental safety of weed biological control agents, provided the insect responds with clear repellency to the plant cues of the tested T&E species. Should this not be the case, additional tests may be warranted. For example, indifferent responses of a biological control agent during bioassays could further be explored in field cages using mixed plots of the weed and the test plant species to test whether indifference would lead to invisibility or whether such plant species would be attacked. Future studies could also consider additional plant cues that biological control agents use during host selection in the field, but that were not included in our behavioral bioassays. These cues may include visual patterns (entire plant), or VOCs of plants with previously damaged or fed upon inflorescences, VOCs that vary through the season, or effects of potential aggregation pheromones of *M. borraginis*.

The data provided here and in *Park et al. (2018)* were included in a petition for field release of *M. borraginis* as a classical biological control agent in the United States, a technical report submitted to the United States Department of Agriculture Animal Plant Health Inspection Service (USDA APHIS), the regulatory agency responsible for biological control release permits in the USA and the Technical Advisory Group (TAG), a committee consisting of members of all federal agencies with a land management mandate that reviews and makes recommendations regarding petitions (M. Schwarzländer, H.L. Hinz, R.L. Winston, 2020, unpublished data; https://www.aphis.usda.gov/plant_ health/permits/tag/downloads/TAGPetitionAction.pdf). Reviewers of TAG commented positively on the data on the pre-alightment host selection of the weevil with regard to T&E plant species. *Mogulones borraginis* was recommended for field release in the United States by TAG in February 29, 2021. The petition for field release is currently undergoing regulatory processes and a release permit is currently expected for 2024. Once the weevil has been released and is established, we recommend monitoring areas in which T&E or otherwise rare or sensitive nontarget species (*e.g., D. daubenmirei*) grow sympatrically with *C. officinale* (see Fig. 1) for any nontarget attack. To further validate the approach described here, field experiments could be conducted in which *M. borraginis* is placed on respective native confamilials at sites where they grow with and sites where they grow without surrounding *C. officinale*, and attack is monitored.

## CONCLUSIONS

We included ecological determinants such as olfactory and visual plant cues for the host recognition of a weed biological control candidate in pre-release environmental safety assessments. The behavioral bioassays and physiological analyses provide an additional layer of pre-release host range assessment data that can improve the reliability of pre-release assessments to predict post-release nontarget attack on T&E plant species (*Havens et al., 2019*; *Heard, 2000*; *Hinz et al., 2014*; *Hinz, Winston & Schwarzländer, 2019*; *Hinz, Winston & Schwarzländer, 2020*; *Schaffner, Smith & Cristofaro, 2018*; *Wheeler & Schaffner, 2013*). Results of our study indicate that the non-destructive collection of olfactory plant cues and subsequent behavioral bioassays can provide a valid (and potentially only) alternative to traditional host-specificity tests in cases where it is impossible to obtain or grow especially important test plants, such as federally listed T&E species. We conclude that studies on pre-alighting host selection behavior and the underlying physiological mechanisms of how biological control organisms select host plants they exploit can aid in environmental safety assessment, particularly of T&E plant species in classical biological weed control. We further argue that such non-destructively collected data on T&E species provides superior host-specificity data compared to that obtained from using surrogate species.

## ACKNOWLEDGEMENTS

We thank William Price for his assistance with statistical analysis. We are grateful to Holly Forbes of the Botanical Garden at the University of California at Berkeley and Cherilyn Burton at Native Plant Program at California Department of Fishery and Wildlife for seeds of *A. grandiflora* and *P. strictus*. We would also like to thank Kelly Amsberry at the Oregon Department of Agriculture for sending us seeds of *P. hirtus* and Wendy Gibble at the University of Washington's Washington Rare Plant Care and Conservation Center for seeds of *H. venusta*. We are also grateful to Lauri Malmquist at USDA Forest Service and Jennifer Andreas at Washington State University for coordinating volatile collection from *H. venusta* in Leavenworth, Washington and Kelly Amsberry and Rob Gibson II for assistance with field work. Finally, we would like to thank Ying Wu at the University of Idaho for assistance with the gas chromatography-mass spectrometry. This is a publication of the Idaho Agricultural Experimental Station.

### Funding

Hariet Hinz and Urs Schaffner were financially supported by CABI with core financial support from its member countries (http://www.cabi.org/about-cabi/who-we-work-with/key-donors/). This research was financially supported by the United States Department of Agriculture Forest Service (Coop. Agreement 10-CA-1142002 to Mark Schwarzländer), the United States Department of Agriculture APHIS CPHST (Agreement 12-8130-1447-CA to Mark Schwarzländer), and the United States Department of Interior Bureau of Land Management (CESU Agreement HAA0807402 to Mark Schwarzländer). The funders had

no role in study design, data collection and analysis, decision to publish, or preparation of the manuscript.

## Grant Disclosures
The following grant information was disclosed by the authors:
CABI.
United States Department of Agriculture Forest Service: 10-CA-1142002.
United States Department of Agriculture APHIS CPHST: 12-8130-1447-CA.
United States Department of Interior Bureau of Land Management: HAA0807402.

## Competing Interests
The authors declare that they have no competing interests.

## Author Contributions
- Ikju Park conceived and designed the experiments, performed the experiments, analyzed the data, prepared figures and/or tables, authored or reviewed drafts of the article, and approved the final draft.
- Mark Schwarzländer conceived and designed the experiments, analyzed the data, authored or reviewed drafts of the article, and approved the final draft.
- Sanford D. Eigenbrode conceived and designed the experiments, analyzed the data, authored or reviewed drafts of the article, and approved the final draft.
- Bradley L. Harmon performed the experiments, prepared figures and/or tables, and approved the final draft.
- Hariet L. Hinz conceived and designed the experiments, authored or reviewed drafts of the article, and approved the final draft.
- Urs Schaffner conceived and designed the experiments, authored or reviewed drafts of the article, and approved the final draft.

## Data Availability
  The raw data for behavioral bioassays are available in the Supplemental File.

## Supplemental Information
Supplemental information for this article can be found online at http://dx.doi.org/10.7717/peerj.16813#supplemental-information.

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
