# Peer review of "Non-destructive environmental safety assessment of threatened and endangered plants in weed biological control"

_PeerJ, doi:10.7717/peerj.16813_

## Round 0.1 · original submission · Minor Revisions

The submitted manuscript is well structured and reflects the main results of long-term field research. The topic of the research is relevant, and the scientific methods used are relatively new, which increases the novelty of the obtained results. The manuscript meets the requirements of the journal, the material is presented clearly. In the process of analyzing the manuscript, the reviewers made a number of comments and recommendations for its overall improvement. I ask the authors of the manuscript to familiarize themselves with them.

Reviewer 1 ·

Basic reporting

• All text is written in refined, straightforward English.
• Very well-written background and introduction sections. The literature cited is current and well-referenced.
• Figures are pertinent, good, properly labeled, and described.
• Raw data provided

Experimental design

Mss is within the journal's scope and includes original primary research.
• The research question is clearly stated, pertinent, and important. The research's contribution to bridging a knowledge gap is described.
• A thorough inquiry that meets the highest technical and ethical standards.
• Methods used are Replicable and are detailed in enough depth.

Validity of the findings

• The researchers working in the field of biological control will definitely get benefited from this research.
• Every piece of supporting information has been offered; it is reliable, statistically sound, and regulated.
• Conclusions are clearly expressed, related to the primary research question, and only include results that support them.

Additional comments

• I applaud the authors for their comprehensive data set, which was assembled over several years of meticulous fieldwork. The manuscript is also written in a crisp, formal style that avoids ambiguity. Mss. is statistically sound. I recommend the Mss for publication.

Reviewer 2 ·

Basic reporting

Overall, this is a well written paper with solid methods and interesting and informative results. The repellent effect of the volatiles from tested non-target plants is quite striking and may have implications for biocontrol testing in the future. There are just a few issues to address and minor corrections to be made.

Experimental design

no comment

Validity of the findings

To firmly conclude that this approach could replace or even contribute to biocontrol safety testing requires showing that there is a correspondence between the response to the isolated volatiles, and the response to whole plants (as traditionally tested). This was not shown. While the tube test approach isolates olfactory responses, it does not include all the cues that an insect may experience from the whole plant, including other volatiles not captured from the cut flowers, volatiles that vary through the season, visual cues of the whole plant (beyond the small piece of plant in the tube), or aggregation pheromones coming from stray individuals that incidentally arrive on the NT plant (including males that were not tested). To best predict what will happen in the field, biocontrol host specificity testing should be as natural and holistic as possible. This shortcoming should be mentioned in the Discussion section.

Related to this, I think that an easier and slightly more natural approach (that is still relatively non-destructive) would be to carry out tests using freshly cut stems held in vials of water that are offered to the insects in cages (either choice or no-choice). While it would not be possible to measure larval development (because the cuttings would not last that long), you could get oviposition results this way and that is more information than you can get from using the volatiles alone. The authors could add some discussion about this alterative approach. Maybe it is still too destructive to take cuttings of T&E plants?

Additional comments

Were there any traditional host range tests performed with M. borraginis on these T&E plant species (or closely related surrogates) in previous studies? If so, what were those results and how did they compare to this more reduced approach?
Although there was a strong repellent effect with this particular group of plants, suggesting the insect is likely to be safe in the field, other biocontrol systems with other groups of test plants may not show such a strong response. Then what? Should decisions be based on a weaker response? Should follow up be done using whole plants if results suggest that a plant species does not repel?
Line 25-27: As stated, it is unclear what the underlying implication is…that these agencies are slow? Non-permissive?
29: Surrogates needs defining here.
30: Replace “were” with “are often”
57: A brief description of the insect’s life history would be helpful, either in the introduction or in methods. This would make clear why you are using flowering stems.
62-64: Sentence unclear and needs rephrasing. Were these plants selected for the current study due to the ESA (as written), or they were listed as T and E because of the ESA, or were they included on the test plant list because of the ESA?
84: Is there a source to cite for the effectiveness of this method?
107: Describe the device a bit more. How long are the arms and what is the diameter of the tubes?
112: How do you know that the ethanol rinse was effective at clearing any volatiles or pheromones in the tube?
123: rather than “first”, I think it should say “next”
125: Need more details. You have not isolated the visual cue since a cut stem will also give off volatiles. The results are similar for both cues (and combined cues) and it seems likely that the insects are responding to olfactory cues in all cases. Also, I would think that the visual cue is very different inside this small tube vs. the visual cue from a whole plant.
Figure 3: What do the error bars represent? Also, rather than labeling as “Behavioral responses…”, I recommend using something that is more descriptive, such as, “Percent of females moving toward the cue”
230: Repelled, not repels
258: I don’t think you can be certain about this. Replace “would not be” with “are very unlikely to be”.
259-261: The study did not test whether the approach adequately predicts field use. You could rephrase to indicate that it is an improvement over not testing these T and E plants at all.
271: Replace “greatly reduces the risks typically associated with pre-release testing” with “eliminates the need to collect T&E plants from the field.” Just trying to make it clearer and more specific.

Reviewer 3 ·

Basic reporting

The paper is well written and clear.

Experimental design

The design is well thought out.

Validity of the findings

Research findings are sound.

Additional comments

This is an interesting paper regarding the use of olfactory cues to determine the likelihood of non-target attack by candidate biological control agents. The testing is sound and the paper worthy of publication.
Since the method is novel, it would have been useful to compare this method using non-T & E species with the classical choice trials, in which candidate agents are also using cues to locate suitable hosts, albeit in a cage environment. For the purpose of comparison, perhaps larger walk-in cages could have been used.
The authors state in the Discussion that Mogulones borraginis was recommended for release in February 2021 but have not clarified whether release was approved. It appears not. It would have also been useful to state status clearly in the Abstract to give weight to the testing method.
Sometimes the author use biocontrol, other times biological control. I think it is best to be consistent. For publications, I think the more formal name: biological control is preferable.
Specific comments
Line 2. Suggest clarifying that the pre-release studies are actually pre-release risk assessment studies, as indicated further on. This is because pre-release studies could also mean studies in the field prior to release, to gauge effectiveness of the agent.
Lines 10-11. Suggest clarifying that this pertains to weed biological control
Lines 13 and 14. Suggest rewording to “weed biological control” on each occasion.
Line 20. I presume the authors mean even if the candidate agent could develop on a NTA in artificial cage trials.
Lines 24-25. Suggest rewording to “weed biological control” and elsewhere in the manuscript.
Line 48. Insert order and family name of the candidate biological agent and the family name of the weed insert a comma after the family name before “with”.
Line 82. Define VOC, as it is the first time it is used.
Line 221. Please spell the name of your organism correctly. Check spelling of all organisms mentioned in the manuscript.
Line 225. Should read D. daubenmirei.
Line 254. Insert fullstop after “M”
Line 268. Replace “4” with “four”
Line 277. Suggest rewording to “weed biological control”
Line 284. Hackelia has already been mentioned previously. Shorten to “H.”
Line 302. Suggest replace “has been” with “was”

---

## Round 0.2 · accepted · Accept

The manuscript has been qualitatively revised and improved. As a result, the conclusions are better substantiated. The manuscript is written with high quality and corresponds to the direction of the journal and its general requirements.